# Controllable Preparation of Superparamagnetic Fe_3_O_4_@La(OH)_3_ Inorganic Polymer for Rapid Adsorption and Separation of Phosphate

**DOI:** 10.3390/polym15010248

**Published:** 2023-01-03

**Authors:** Yao Lu, Xuna Jin, Xiang Li, Minpeng Liu, Baolei Liu, Xiaodan Zeng, Jie Chen, Zhigang Liu, Shihua Yu, Yucheng Xu

**Affiliations:** 1Jilin Institute of Chemical Technology, College of Chemical & Pharmaceutical Engineering, Jilin 132022, China; 2Jilin Institute of Chemical Technology, School of Petrochemical Technology, Jilin 132022, China; 3Jilin Institute of Chemical Technology, Centre of Analysis and Measurement, Jilin 132022, China; 4Railway Transportation Department, Jilin Petrochemical Company, Jilin 132021, China

**Keywords:** superparamagnetic, Fe_3_O_4_@La(OH)_3_ inorganic polymer, phosphate adsorption

## Abstract

Superparamagnetic Fe_3_O_4_ particles have been synthesized by solvothermal method, and a layer of dense silica sol polymer is coated on the surface prepared by sol-gel technique; then La(OH)_3_ covered the surface of silica sol polymer in an irregular shape by controlled in situ growth technology. These magnetic materials are characterized by TEM, FT-IR, XRD, SEM, EDS and VSM; the results show that La(OH)_3_ nanoparticles have successfully modified on Fe_3_O_4_ surface. The prepared Fe_3_O_4_@La(OH)_3_ inorganic polymer has been used as adsorbent to remove phosphate efficiently. The effects of solution pH, adsorbent dosage and co-existing ions on phosphate removal are investigated. Moreover, the adsorption kinetic equation and isothermal model are used to describe the adsorption performance of Fe_3_O_4_@La(OH)_3_. It was observed that Fe_3_O_4_@La(OH)_3_ exhibits a fast equilibrium time of 20 min, high phosphate removal rate (>95.7%), high sorption capacity of 63.72 mgP/g, excellent selectivity for phosphate in the presence of competing ions, under the conditions of phosphate concentration 30 mgP/L, pH = 7, adsorbent dose 0.6 g/L and room temperature. The phosphate adsorption process by Fe_3_O_4_@La(OH)_3_ is best described by the pseudo-second-order equation and Langmuir isotherm model. Furthermore, the real samples and reusability experiment indicate that Fe_3_O_4_@La(OH)_3_ could be regenerated after desorption, and 92.78% phosphate removing remained after five cycles. Therefore, La(OH)_3_ nanoparticles deposited on the surface of monodisperse Fe_3_O_4_ microspheres have been synthesized for the first time by a controlled in-situ growth method. Experiments have proved that Fe_3_O_4_@La(OH)_3_ particles with fast separability, large adsorption capacity and easy reusability can be used as a promising material in the treatment of phosphate wastewater or organic pollutants containing phosphoric acid functional group.

## 1. Introduction

Phosphorus (P) is one of the indispensable nutrient elements for the growth and reproduction of all organisms. However, the excessive discharge of phosphate not only causes the eutrophication of aquatic ecosystems but also aggravates the shortage of global phosphorus resources, making the global food crisis even worse. Therefore, it is urgent to develop an effective method to remove and recover excessive phosphate from wastewater for sustainable development [1].

Numerous treatment technologies, such as electrocoagulation [2], crystallization [3], biological treatment [4], chemical precipitation [5], membrane separation [6], ion exchange [7] and adsorption [8,9] have been developed to remove phosphate from wastewater. Among them, adsorption is considered as a promising technology for adsorbate effective removal and recovery. At present, a lot of functionalized adsorbents have been utilized for phosphate elimination, including cation exchange resin [10], layered double hydroxides [11], fibrous adsorbent [12], hydrogel [13], metal organic frameworks [14], affinity membrane [15] and porous biochar [16]. Owing to the strong affinity between La and phosphate, La-based materials have attracted much attention. In particular, lanthanum (hydr)oxide offers both pore structures and large specific surface areas and provides a large number of coordination sites towards phosphate even at low concentration [17,18].

To enhance separation and re-dispersion performance of materials, Fe_3_O_4_ is used as magnetic core to construct composites with the magnetic property and specific adsorption [19]. In recent years, it has been reported in some literatures that various magnetic particles have been used to remove phosphate, achieving good results [20,21,22,23,24]. However, these materials either have small specific surface area or have poor dispersion due to irregular shape and uneven size, which limits the contact with the adsorbate to some extent and reduces the removal efficiency of magnetic particles on phosphate.

In the early stage, we realized the controllable preparation of monodisperse and superparamagnetic Fe_3_O_4_ magnetic beads with diameter of 50, 200, 400 nm [25,26,27,28,29,30,31]. In this paper, La(OH)_3_ nanoparticles are grown in situ on the surface of the above functional magnetic beads and successfully prepared Fe_3_O_4_@La(OH)_3_ inorganic polymer. In the past, La(OH)_3_/Fe_3_O_4_ composite has been established to act as an adsorbent for phosphate, with large adsorption capacity, and can be separated by magnetic separation [32]. However, Fe_3_O_4_ used in most literatures is prepared by coprecipitation method. In comparison to coprecipitation method, Fe_3_O_4_ is prepared by solvent method in this study with the characteristic of monodisperse and fast magnetic response, and La(OH)_3_ can be directly modified on its surface. In addition, since Fe_3_O_4_ has no dense protective layer, it is easy to be oxidized, and the corresponding magnetic response is weakened, so magnetic separation recycling cannot be achieved. We can improve the stability of Fe_3_O_4_ by modifying its surface with a layer of dense silica and then with La(OH)_3_. A combination of the advantages described above with solid phase extraction spectrometry and the adsorption properties of Fe_3_O_4_@La(OH)_3_ for phosphate (adsorption performance optimization, kinetic, isotherm) is evaluated using batch experimental methods. Besides, actual sample testing and Fe_3_O_4_@La(OH)_3_ composite reusability are also investigated in this study.

## 2. Materials and Methods

### 2.1. Materials

Iron chloride hexahydrate, poly(ethylene glycol), ethylene glycol were purchased from Aladdin Biochemical Technology Co., Ltd. (Shanghai, China); Sodium acetate (AR, ≥99%), ascorbic acid (AR, ≥99.7%), potassium dihydrogen phosphate (KH_2_PO_4_, AR, ≥99%), potassium antimony tartrate (AR, ≥99%), ammonium molybdate (AR, ≥99%), sulfate triacetyloxysilyl acetate (AR, ≥99%), lanthanum nitrate hexahydrate (AR, ≥99%) and ethanol were obtained from Sinopharm Chemical Reagent Co., Ltd. (Beijing, China).

Accurately, a certain mass of KH_2_PO_4_ was dissolved with water as phosphate stock solution, and the concentration was 1000 mgP/L. The adsorbed solution was prepared by diluting the stock solution with distilled water.

### 2.2. Preparation of Adsorbent

#### 2.2.1. Synthesis of Fe_3_O_4_ Particles

First, 60 mL ethylene glycol and 1.0 g polyethylene glycol were added into a 100 mL beaker and stirred in a 65 °C water bath until they were dissolved. Then 2.7 g of ferric chloride hexahydrate was dissolved in the above solution. After complete dissolution, 7.2 g of sodium acetate trihydrate was added and stirred for another 20 min. Finally, the solution was transferred to 100 mL polytetrafluoroethylene (PTFE) lined hydrothermal autoclave reactor and placed in a vacuum oven at 200 °C to react for 10 h. The black products were obtained and separated from the mixture by a magnet. Then they were washed with water and alcohol and dried in an oven at 60 °C.

#### 2.2.2. Synthesis of Fe_3_O_4_@SiO_2_ Composite

The as-prepared Fe_3_O_4_ particles (0.1 g), absolute ethanol (48 mL) and distilled water (12 mL) were dispersed in a 3-necked flask under ultrasonication for 10 min. NH_3_∙H_2_O (2 mL) was added in the above solution and stirred mechanically for 20 min. A mixed solution of TEOS (1 mL) and absolute ethanol (1 mL) was slowly dropped into the 3-mouth flask and reacted at room temperature for 5 h. After complete reaction, the product was washed with water and alcohol; the final product was diluted with water to a concentration of 50 mg/mL.

#### 2.2.3. Synthesis of Fe_3_O_4_@La(OH)_3_ Composites

The above prepared Fe_3_O_4_@SiO_2_ (10 mL), distilled water (40 mL) and NaOH solution (10 mL, 0.5 M) were added into a three-mouth flask and stirred for 60 min. Then, La(NO_3_)_3_ (0.1 g) that dissolved it with 50 mL distilled water was slowly dropped into the above solution (30 min), and stirring continued for 4 h. Finally, the supernatant was poured out and washed with water for 5 times with the help of magnetic separation.

### 2.3. Characterization of Adsorbent

Transmission electron microscopy (TEM) images were investigated using a JEOL JEM-2100F TEM (Tokyo, Japan). Fourier transform infrared spectra (FT-IR) were performed on a Spectrum One (PerkinElmer, Shelton, CT; USA) instrument to determine the functional groups of the absorbents. Scanning electron microscopic (SEM) image was obtained on a Hitachi S-4800 SEM (Tokyo, Japan), and the element mapping was analyzed on an Energy Dispersive X-Ray Spectroscopy (EDX) detector. The crystal structure of the particles was characterized by a D8FOCUS X-ray diffractometer (Bruker, Germany) with Cu Kα radiation (k = 1.5406 Å). An UV-Vis T6 spectrometer (Beijing, China) was used to measure the concentration of phosphate. The pH values of solution were tested by Sartorius PB 220 pH meter (Gottingen, Germany).

### 2.4. Adsorption Experiments

The adsorption experiments were performed in triplicate, and the phosphate solutions used were prepared with standard phosphate solutions. The adsorption capability and removal efficiency of phosphate on Fe_3_O_4_@La(OH)_3_ was calculated by the following equation:(1)$$q_{e}=\frac{\left(C_{0}-C_{e}\right)\times V}{m}$$
(2)$$R_{e}=\frac{C_{0}-C_{e}}{C_{0}}\times 100\%$$
where R_e_ (%) and q_e_ (mg/g) are the removal efficiency and adsorption capacity of phosphate, respectively. C_e_ and C_0_ (mgP/L) represented the equilibrium and initial phosphate concentrations, respectively. V (L) is the volume of phosphate solution, and M (g) is the weight of Fe_3_O_4_@La(OH)_3_.

The adsorption performance of Fe_3_O_4_@La(OH)_3_ composites for phosphate were investigated, by adding different doses of adsorbent (5–40) mg to a 3-mouth flask filled with phosphate solution (50 mL) and mechanically stirred at room temperature. When the adsorption experiment reached equilibrium, then magnetic separation, the residual phosphate concentration in the solution was tested. To study the change of pH, 30 mg Fe_3_O_4_@La(OH)_3_ were dispersed in 50 mL of phosphate solution (30 mgP/L) at 25 °C. Solution pH change from 3.0 to 11.0 was adjusted with a 0.1 M HCl or 0.1 M NaOH solution. The influence of common coexisting anions on phosphate sorption were examined by adding Cl^−^, CO_3_^2−^, SO_4_^2−^, Na^+^ and K^+^ to the phosphate solutions at room temperature.

### 2.5. Adsorption Kinetics

To study the adsorption kinetics, 30 mg of adsorbent was dispersed in a 3-mouth flask containing 50 mL of phosphate with an initial concentration of 30 mgP/L and then stirred mechanically, and samples were taken with a pipette gun for certain time intervals at room temperature.

### 2.6. Adsorption Isotherm

Sorption isotherm experiments were completed in an 18–38 °C water bath; 30 mg of Fe_3_O_4_/La(OH)_3_ were dispersed into phosphate solution (50 mL) with initial concentration ranging from 5 to 100 mgP/L, and the maximum adsorption amount was investigated.

### 2.7. Reusable Performance

The saturated adsorbent was magnetically separated and then dispersed in 1.0 M NaOH solution for phosphate desorption until the equilibrium state. The regenerated Fe_3_O_4_@La(OH)_3_ was washed with deionized water and then reused in the succeeding cycle.

### 2.8. Real Samples

To demonstrate the applicability of Fe_3_O_4_@La(OH)_3_, extra adsorption experiments were carried out in Tap water and Songhua River water, respectively. Water samples were filtered with microporous membrane, then spiked phosphate to 30 mg/L by standard solutions.

## 3. Results and Discussion

### 3.1. Preparation of Fe_3_O_4_@La(OH)_3_ Composite

The fabrication procedure for the Fe_3_O_4_@La(OH)_3_ is shown in Figure 1. From the scheme, (i) super-paramagnetic Fe_3_O_4_ particles are synthesized by solvothermal reaction as the magnetic core of composite, making it convenient to separate from the matrix by a magnet. (ii) The as-prepared Fe_3_O_4_ particles are coating with silica by sol-gel method to increase its water solubility and monodispersity. (iii) Then the surface of Fe_3_O_4_@SiO_2_ is covered with La(OH)_3_ in an irregular shape via an in situ reduction reaction [32,33]. Meanwhile, TEM images of the magnetic particles obtained at each stage are also given in Figure 1, which proves the successful synthesis of Fe_3_O_4_@La(OH)_3_ composite and indicates that the surface of Fe_3_O_4_ is modified by La(OH)_3_ particle in the composite, compared to the relatively smooth surface of Fe_3_O_4_.

### 3.2. Characterization of Fe_3_O_4_@La(OH)_3_ Composite

A suite of techniques, including FT-IR, XRD, VSM and SEM, are used to characterize the Fe_3_O_4_@La(OH)_3_ composite. In the FT-IR spectra (Figure 2A), the relatively high intensity of a band at 580 cm^−1^ indicates the stretch of Fe–O [34]. In addition, the peaks at 795 cm^−1^ and 472 cm^−1^ could be assigned to the symmetrical stretching vibration of Si–O. The strong and broad absorption band near 1090 cm^−1^ is consistent with the characteristic absorption of Si–O–Si antisymmetric stretching vibration [35]. There is no obvious difference in absorption peaks between Fe_3_O_4_@SiO_2_ and Fe_3_O_4_@La(OH)_3_, mainly because the La–O and Si–O bond absorption peaks are overlapped [36]. The crystalline structures of magnetic particles are demonstrated by XRD as shown in Figure 2B. The positions of all diffraction peaks matched the JCPDS card (75–1610) for magnetite perfectly [37], which suggested that the introduction of SiO_2_ and La(OH)_3_ had no effect on the response of Fe_3_O_4_ signals. In the meantime, the intensities of the main peaks have decreased marginally, which reveals that the SiO_2_ layer is introduced and La(OH)_3_ is successfully loaded. Figure 2C shows the magnetization curves of magnetic composite; it is discovered that the saturation magnetization of Fe_3_O_4_@La(OH)_3_ is decreased compared with that of Fe_3_O_4_@SiO_2_. That is because the shielding effect of La(OH)_3_ introduced on the surface of Fe_3_O_4_, and the reduction of the relative content of Fe_3_O_4_ in the composite. Meanwhile, Fe_3_O_4_@La(OH)_3_ still show good monodispersity, strong magnetism and rapid separation by using an external magnetic field (insert picture).

The microstructure and morphology of the magnetic particles are characterized using scanning electron microscopy (SEM). Figure 3A shows the magnetic cores have uniform size and rough surface. As is shown in Figure 3B, the surface of particles become smooth and flat due to the coating of nano layer, which further proves that the silica sol polymer is successfully prepared by sol-gel method. After the deposition of La(OH)_3_ (Figure 3C), the Fe_3_O_4_@SiO_2_ particles are almost wrapped into the target material. Thus, the formation of the composite resulted in the increase of size, and the average diameter of Fe_3_O_4_@La(OH)_3_ is about 420 nm (Figure 3B), compared with Fe_3_O_4_@SiO_2_ that has an average diameter of 370 nm (Figure 3A). Based on the SEM images, we can see magnetic composites are homogeneous, monodisperse and superparamagnetic particles. In addition, elemental analysis confirms the Fe_3_O_4_@La(OH)_3_ successfully prepared, as Fe, Si, O and La elements exist throughout the material matrix (Figure 3D). Then EDX elemental mapping of Fe_3_O_4_@La(OH)_3_ has been presented in Appendix A, verifying the successful formation of a core-shell structure in the composite and the contents of Fe, Si and La are 90.15%, 5.99% and 3.85%, respectively.

### 3.3. Optimization of Adsorption Experiment

#### 3.3.1. Batch Factors Experiments

The adsorbent dosing levels on phosphate adsorption capacity and removal rate are tested to select the appropriate amount of magnetic lanthanum hydroxide. As presented in Figure 4A, phosphate adsorption capacity q_e_ decreased from 97.94 mg/g to 35.20 mg/g, with an increasing of the dosage from 5–40 mg. This result shows that less dosage is accompanied by less adsorption sites, resulting in saturation of adsorption sites and high adsorption capacity. With the increase of adsorbent dosage, the surface adsorption sites increase, while adsorption cannot reach saturation since the phosphate concentration is constant. Meanwhile, the phosphate removal rate gradually increased from 32.63% to 96.87% with the increase of dosage. On the other hand, the rate of phosphate removal tends to be stable at the adsorbent dosage of 30 mg. Considering the sufficient adsorption and high removal efficiency, the adsorbent dosage of 30 mg is adopted in the subsequent experiment.

An experimental study on pH is carried out in the ranges of pH from 2 to 9. As can be seen from Figure 4B, phosphate adsorption on Fe_3_O_4_@La(OH)_3_ is greatly affected by the pH value of solution. As we know, H_2_PO_4_^−^ and HPO_4_^2−^ are the dominant species of phosphate in the solution in the investigated pH range. At the same time, the surface charges of Fe_3_O_4_@La(OH)_3_ have been tested at different pH values by the zeta potential measurement (Appendix A), and its isoelectric point is found to be 4.89. Thus, when pH < 4.89, the surface of Fe_3_O_4_@La(OH)_3_ particles became positively charged; the phosphate anions are favorable adsorbed because of the electrostatic attraction. Meanwhile, as the pH > 4.89, Fe_3_O_4_@La(OH)_3_ is negatively charged; electrostatic repulsion leads to the decrease of phosphate adsorption capacity; and the phosphate removal decreased with the increase of pH [38,39]. For all that, a high phosphate removal rate (>95.7%) is maintained in the pH = 3–7 range.

#### 3.3.2. Influence of Interfering Ions

As the phosphate adsorption on Fe_3_O_4_@La(OH)_3_ surfaces may interfere with common coexisting ions in environmental water, some interfering ions including SO_4_^2−^, Cl^−^, CO_3_^2−^, Na^+^ and K^+^ on phosphate enrichment at pH 7.0 are investigated. As evidenced in Figure 5, these foreign ions do not cause significant interference, even when the concentration of the competing ions is at 10 times (300 mg/L) excess.

### 3.4. Adsorption Kinetics

The kinetic data obtained are analyzed by using pseudo-first-order rate (Equation (3)) and pseudo-second-order rate (Equation (4)), respectively [40]. A comparison of the kinetic models for phosphate adsorption by Fe_3_O_4_@La(OH)_3_ using nonlinear regression is presented in Figure 6, and the relevant results are summarized in Table 1.

Pseudo-first-order kinetic model:(3)$$\ln\left(q_{e}-q_{t}\right)={\mathrm{lnq}}_{e}-k_{1}t$$

Pseudo-second-order kinetic model:(4)$$\frac{t}{q_{t}}=\frac{1}{k_{2}q_{e}^{2}}+\frac{t}{q_{e}}$$
where q_t_ (mg/g) is phosphate adsorption (mg/g) at time t and q_e_ (mg/g) is the equilibrium adsorption; k_1_ (min^−l^) and k_2_ (g/(mg·min)) were the rate constants of pseudo-first-order and pseudo-second-order reaction, respectively.

As shown in Figure 6 and Table 1, the second-order kinetic model fit all the adsorption data well according to the relatively high correlation coefficients (r > 0.99). Moreover, the experimental date (q_e,exp_) is consistent with the calculated date (q_e,cal_) obtained from pseudo-second-order model. The results indicating that the rate limiting step might be due to chemical absorption, and its adsorption capacity is directly proportional to the number of surface active binding sites of magnetic materials.

### 3.5. Adsorption Isotherms

Langmuir and Freundlich models are used to analyze the adsorption data [41]. The fitting equations are expressed as follows:(5)$$\frac{C_{e}}{q_{e}}=\frac{1}{k_{L}q_{\max}}+\frac{C_{e}}{q_{\max}}$$
(6)$${\mathrm{lnq}}_{e}=\frac{1}{n}{\mathrm{lnC}}_{e}-{\mathrm{lnk}}_{F}$$
where C_e_ (mg/L) is equilibrium concentration; q_max_ (mg/g) is the highest adsorption capacity; n is a parameter that represents the medium heterogeneity and adsorption intensity; and k_L_ (L/mg) and k_F_ ((mg/g)/(mg/L)^1/n^) are Langmuir and Freundlich constants, respectively. The fitting curves and correlation parameters are shown in Table 2.

From the correlation coefficient (R^2^) of two models, it is discovered that the Langmuir model is more reasonable to interpret the isotherms of the adsorption process, suggesting that phosphates are absorbed in monolayer style, and the maximum absorption value of q_max_ predicted is 105.37 mg/g.

### 3.6. Removal of Phosphate in Actual Water and Its Reusability

The prepared Fe_3_O_4_@La(OH)_3_ (30 mg) is added into 50 mL tap water and Songhua River water, respectively. None of the original samples are evaluated to contain phosphate as tested by solid phase extraction spectrometry and then two samples are spiked phosphate to 30 mg/L with standard solutions for subsequent extraction and detection. The results of actual water samples detection and composite particles reusability are shown in Figure 7. Figure 7A reveals that the adsorption rates of Fe_3_O_4_@La(OH)_3_ for phosphate in tap water and Songhua water are 96.17% and 95.46%, demonstrating the practical application of phosphate removal on Fe_3_O_4_@La(OH)_3_ would be applicable. Also, the reusability of the adsorbent for phosphate is illustrated in Figure 7B; the phosphate removal rate could still reach 92.78% after five cycles of reuse [42]. The adsorption capacity of Fe_3_O_4_@La(OH)_3_ in our study is also contrasted with the materials reported in the literatures (Appendix A); the comparison results show Fe_3_O_4_@La(OH)_3_ particles exhibit significant advantages to realize rapid and efficient phosphate removal.

### 3.7. Mechanism of Phosphate Removing by Fe_3_O_4_@La(OH)_3_

To gain insight into the mechanisms of phosphate removing by the Fe_3_O_4_@La(OH)_3_ composite, the zeta potential measurements have been performed (Appendix A shows that the isoelectric point (pH_zpc_) of Fe_3_O_4_@La(OH)_3_ particles is found to be about 4.89 before adsorption, while the value decreased to about 4.25 after adsorption. pH_zpc_ moves to a lower pH value due to the accumulation of negative charges within the shear plane, is typically considered as inner-sphere complexation phenomena caused by ligand exchange, which can be described in the following reactions [43,44,45,46]:La-OH + H_2_PO_4_^−^ ↔ La-H_2_PO_4_ + OH^−^
La=OH + HPO_4_^2−^ ↔ La=HPO_4_ + 2OH^−^
La≡OH + PO_4_^3−^ ↔ La≡PO_4_ + 3OH^−^

The involvement of ligand exchange is further evidenced by the observed change of solution pH before and after adsorption. The initial pH of 30 mgP/L phosphate solution is 5.28, and the pH value gradually increased to 7.08 after adsorption for 1 h. The increase in pH value during sorption is attributed to the release of OH^−^ upon ligand phosphate removing. Based on the above analysis, the replacement of the surface hydroxyl groups by phosphate ions with the formation of inner-sphere complex played an important role in the removal of phosphate by Fe_3_O_4_@La(OH)_3_ particles.

## 4. Conclusions

In this study, hydrothermal methods and in-situ precipitation are adopted to prepare Fe_3_O_4_@La(OH)_3_ inorganic polymer, which is confirmed based on the results of TEM, FT-IR, XRD, SEM and EDS. Batch experiments are used to investigate the ability of Fe_3_O_4_@La(OH)_3_ for phosphate removing from water. It is observed that Fe_3_O_4_@La(OH)_3_ composites exhibit a fast equilibrium time of 20 min, high phosphate removal rate (>95.7%), high sorption capacity of 63.72 mgP/g, under the conditions of phosphate concentration 50 mgP/L, pH = 7, adsorbent dose 0.6 g/L and room temperature. Fe_3_O_4_@La(OH)_3_ displayed excellent selectivity and anti-interference toward phosphate over other interfering ions including SO_4_^2−^, Cl^−^, CO_3_^2−^, Na^+^ and K^+^, even if their concentration is 10 times higher than that of phosphate. In addition, the kinetic data and isotherm data are better described by pseudo-second-order equation and Langmuir model, respectively, indicating that monolayer chemisorption occurred during adsorption process. In summary, the Fe_3_O_4_@La(OH)_3_ composites with the characteristics of superparamagnetic, fast adsorption efficiency, easy reusability, etc. will be a promising adsorbent for phosphate removal from wastewater or polluted surface water.

## Figures and Tables

**Figure 1 polymers-15-00248-f001:**
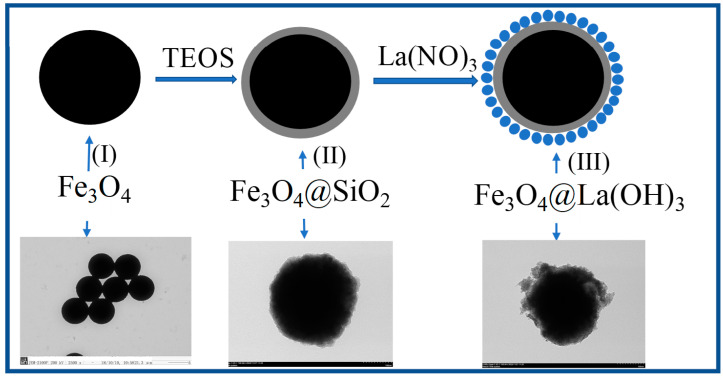
Schematic illustrations of the fabrication procedure for the Fe_3_O_4_@La(OH)_3_.

**Figure 2 polymers-15-00248-f002:**
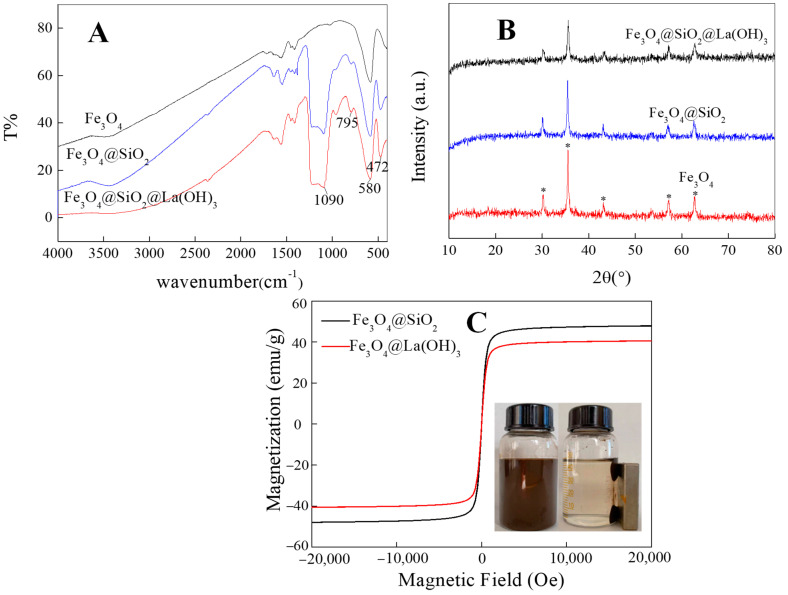
(**A**) FT-IR spectra, (**B**) XRD patterns and (**C**) VSM diagram of magnetic composites.

**Figure 3 polymers-15-00248-f003:**
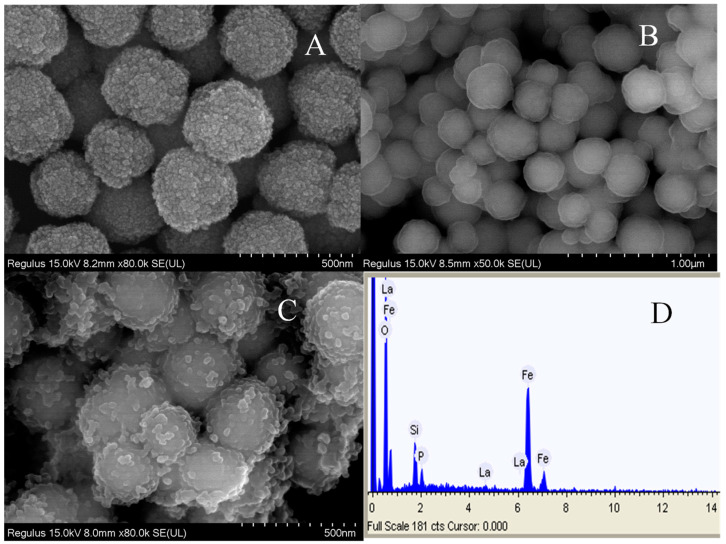
SEM images of (**A**) Fe_3_O_4_, (**B**) Fe_3_O_4_@SiO_2_, (**C**) Fe_3_O_4_@La(OH)_3_ and (**D**) elemental analysis of Fe_3_O_4_@La(OH)_3_.

**Figure 4 polymers-15-00248-f004:**
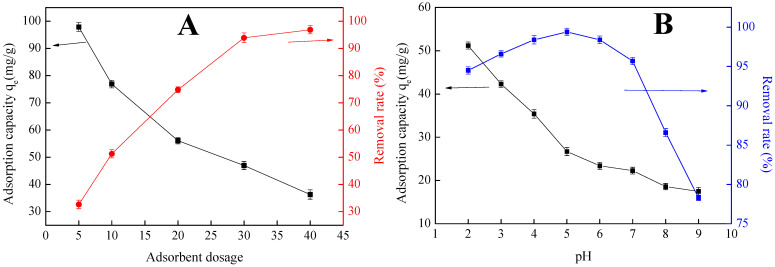
The effect of (**A**) adsorbent dosage and (**B**) the solution pH on phosphate adsorption capacity.

**Figure 5 polymers-15-00248-f005:**
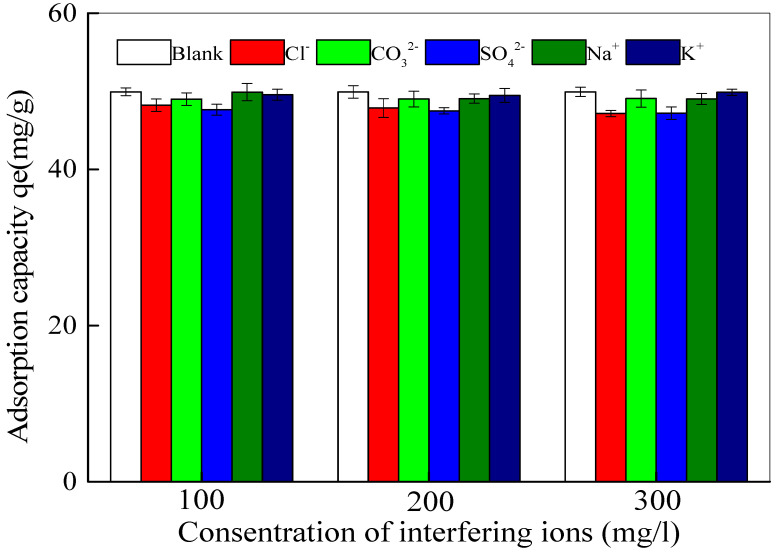
Adsorption capacity for phosphate in the presence of 30 mg/L phosphate and various concentrations of possible interference ions.

**Figure 6 polymers-15-00248-f006:**
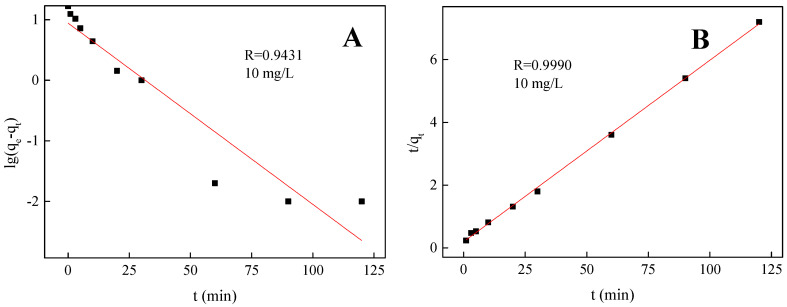
Kinetic curves for phosphate adsorption by Fe_3_O_4_@La(OH)_3_ (**A**) pseudo-first-order (**B**) pseudo-second-order model with initial phosphate concentration of 10 mg/L.

**Figure 7 polymers-15-00248-f007:**
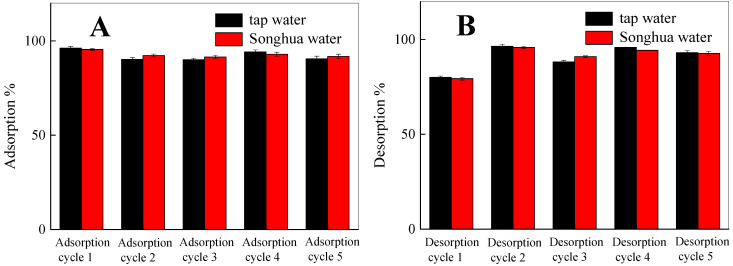
Reusability tests of Fe_3_O_4_@La(OH)_3_ for phosphate removing: five adsorption (**A**) and desorption (**B**) cycles in actual water samples.

**Table 1 polymers-15-00248-t001:** Kinetic parameters for the adsorption of phosphate by Fe_3_O_4_@La(OH)_3_.

C_0_ (mg/L)	q_e,exp_ (mg/g)	Pseudo-First-Order			Pseudo-Second-Order		
		k_1_ × 10^−2^	q_e,cal_ (mg/g)	R^2^	k_2_ × 10^−3^	q_e,cal_ (mg/g)	R^2^
5	8.33	5.73	5.19	0.9236	3.09	8.64	0.9976
10	16.67	6.89	8.77	0.9015	1.68	17.29	0.9990
30	49.93	6.70	28.80	0.8681	0.32	53.16	0.9963
50	61.32	5.83	44.68	0.9146	0.25	65.44	0.9972
100	86.47	8.01	68.91	0.9429	0.13	94.43	0.9951

**Table 2 polymers-15-00248-t002:** Freundlich and Langmuir constants and correlation coefficients for adsorption of phosphate onto Fe_3_O_4_@La(OH)_3_.

Temperature (K)	Langmuir				Freundlich	
	Q_max_ (mg/g)	K_L_ (L/mg)	R^2^	n	K_F_ (mg/g)	R^2^
298	70.77	2.26	0.9994	1.22	0.027	0.9425
318	82.37	1.89	0.9966	1.93	0.026	0.9426
338	105.37	4.07	0.9990	4.34	0.017	0.9059

## Data Availability

All data included in this study are available upon request by contact with the corresponding author.

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
