# Peer review of "Controllable Preparation of Superparamagnetic Fe3O4@La(OH)3 Inorganic Polymer for Rapid Adsorption and Separation of Phosphate"

_polymers, 2023, doi:10.3390/polym15010248_

Round 1
Reviewer 1 Report
Yao Lu et al. Describe in their manuscript a three step synthesis to obtain defined Fe3O4 nanoparticles equipped with a SiO2 shell on which a layer of La(OH)3 species could be anchored. These particles can be used to remove phosphate from waste water by adsorption and can easily be separated due to the magnetic nature of the Fe3O4 core of the particles.
They give a satisfactory report about the solvothermal and sol-gel synthesis, the used characterization methods, kinetic experiments of the adsorption and a short removal experiment using "real" water.
Despite this, I was not able to identify a property of the material or its performance which shows significant improvements compared to the materials published in the last years. It would therefore be helpful to place the results presented here in the context of the current literature on this topic. In the current version, I can not suggest a publication.
Reviewer 2 Report
The authors presenting their findings on magnetic sorbents based on core-shell systems should carefully consider the following issues for their paper to be of interest.
1) The English language needs to be completely revised, it makes reading the manuscript difficult;
2) The introduction should focus more on phosphate adsorption. Magnetic nanoparticles for other application can be interesting but are not the focus of the manuscript. Please add appropriate references on phosphate removal and proposed mechanisms in the literature. Neither in the abstract nor in the Introduction it is made clear what is the improvement brought by these materials to this application;
3) In the materials and methods the type of reactor used in the solvothermal synthesis should be specified;
4) The characterization of the new materials is quite poor. I don't see the point in reproducing twice the same SEM images (Scheme 1 and Figure 3) and not have a TEM image to show the intimate structure of the magnetic material, of the shell, and the Lanthanum hidroxide;
5) Due to their size, the magnetic cores are probably made of aggregated smaller nanoparticles (which can be superparamagnetic). The term "monodisperse" is abused and quite not right, unless proved by a proper size analysis of the produced materials. The few pictures given (the repeated SEM images) don't give a clear idea on the aggregation/coalescence state of the particles after sol-gel synthesis. The composition should be provided, in particular the amount of La per mass of adsorbent. No porosity data have been provided;
6) The results should be properly discussed compared to the literature.